# Effect of Different Edible *Trichosanthes* Germplasm on Its Seed Oil to Enhance Antioxidant and Anti-Aging Activity in *Caenorhabditis elegans*

**DOI:** 10.3390/foods13030503

**Published:** 2024-02-05

**Authors:** Wenqian Wang, Shan Li, Yunguo Zhu, Ruilin Zhu, Xiling Du, Xianghuan Cui, Hongbing Wang, Zhou Cheng

**Affiliations:** School of Life Sciences and Technology, Tongji University, Shanghai 200092, China; 2110808@tongji.edu.cn (W.W.); lishanbio@tongji.edu.cn (S.L.); ygzhu@tongji.edu.cn (Y.Z.); 2231508@tongji.edu.cn (R.Z.); duxiling@tongji.edu.cn (X.D.); cuixh@tongji.edu.cn (X.C.); hbwang@tongji.edu.cn (H.W.)

**Keywords:** edible *Trichosanthes* germplasm, seed oil, fatty acid, antioxidant activity, anti-aging

## Abstract

The seeds of various *Trichosanthes* L. plants have been frequently used as snacks instead of for traditional medicinal purposes in China. However, there is still a need to identify the species based on seeds from *Trichosanthes* germplasm for the potential biological activities of their seed oil. In this study, 18 edible *Trichosanthes* germplasm from three species were identified and distinguished at a species level using a combination of seed morphological and microscopic characteristics and nrDNA-ITS sequences. Seed oil from the edible *Trichosanthes* germplasm significantly enhanced oxidative stress tolerance, extended lifespan, delayed aging, and improved healthspan in *Caenorhabditis elegans*. The antioxidant activity of the seed oil exhibits a significant positive correlation with its total unsaturated fatty acid content among the 18 edible *Trichosanthes* germplasm, suggesting a genetic basis for this trait. The biological activities of seed oil varied among species, with *T. kirilowii* Maxim. and *T. rosthornii* Harms showing stronger effects than *T. laceribractea* Hayata.

## 1. Introduction

*Trichosanthes* L. is a large genus in the Cucurbitaceae family, which includes vital medicinal and edible plant resources. It includes 84 species and eight variants with a global distribution, mainly in eastern and southern Asia and northern Australia [1]. Notably, China harbors 41 species and eight variants, predominantly distributed in the southern and southwestern regions, with cultivation extending to the north and coastal provinces, such as Shandong and Zhejiang. Traditionally, the dried mature fruits, seeds, peels, and roots of *T. kirilowii* Maxim. and *T. rosthornii* Harms can be used in Chinese traditional medicine [2]. Beyond their medicinal values, their seeds with a refreshing taste are also recognized for their high edible value and richness in fat and protein [3]. However, the research on *Trichosanthes* L. plants has mainly focused on traditional uses, such as the phytochemistry and pharmacology of their fruits and roots [4], with limited studies on the seeds of *Trichosanthes* L. plants, particularly regarding their edible uses.

*T. laceribractea* Hayata has a longstanding history of cultivation for edible seed consumption in China [5]. Nowadays, the seeds of various *Trichosanthes* L. plants, including but not limited to *T. kirilowii* Maxim. or *T. rosthornii* Harms, have gained popularity as snack foods following frying and other processing methods [6]. However, the emerging market for these various seeds has resulted in confusion about the edible *Trichosanthes* germplasm, leading to unclear species classification. Morphological characteristics are the most direct indications, and they are usually used to identify plants [7]. Additionally, the microscopic characteristics of the seed coat are relatively stable and have significant value for the genetic classification of plants [8]. With the development of molecular biology, nrDNA-ITS has also been commonly used for species identification and phylogenetic analysis in plants [9]. Therefore, in this study, the seed morphological and microscopic characteristics combined with nrDNA-ITS sequences were studied to reveal the species classification of edible *Trichosanthes* germplasm. The results may be vital for further efficacy evaluation of such *Trichosanthes* germplasm.

The seeds of *T. kirilowii* Maxim. are rich in unsaturated fatty acids (UFAs) and particularly abundant in polyunsaturated fatty acids (PUFAs) [10]. They are also known for numerous physiological functions and health benefits; for example, the components of biological membranes are the precursors of various molecules that participate in the process of energy storage [11]. The intake of PUFAs plays an important role in the proper functioning of the human body and the prevention of cardiovascular diseases, heart attacks, and inflammatory diseases [12]. The seed oil from *T. kirilowii* Maxim. has been reported to demonstrate in vitro antioxidant activity against the superoxide anion radical [13]. However, the variability in fatty acid composition and the potential biological activity of seed oil among different edible *Trichosanthes* germplasm remain unclear.

With the changing lifestyle and the aging global population in modern society, neurodegenerative disorders pose a serious threat to human health. Aging is a major risk factor for most neurodegenerative diseases, including Alzheimer’s disease [14]. The reactive oxygen species (ROS) produced during the metabolic process is one of the main causes of aging [15]. Hence, there is a growing need to find compounds with antioxidant properties, low toxicity, and minimal side effects to combat aging. In this regard, seed oil from edible *Trichosanthes* L. plants shows potential in this area. This study’s objectives were (i) to reveal the species classification of edible *Trichosanthes* germplasm based on seed morphological and microscopic characteristics combined with nrDNA-ITS sequence; (ii) to analyze the yield and fatty acid composition of seed oil from the edible *Trichosanthes* germplasm; and (iii) to evaluate the antioxidant and anti-aging properties of the seed oil using *C. elegans*, aiming to elucidate the causes of differences among the edible *Trichosanthes* germplasm. The findings are expected to provide a scientific foundation for the development and utilization of seeds of *Trichosanthes* L. plants, with potential implications for delaying aging and the prevention of diseases associated with aging.

## 2. Materials and Methods

### 2.1. Plant Materials

Seeds from 18 edible *Trichosanthes* germplasm were collected, including 6 *T. laceribractea* Hayata, 11 *T. kirilowii* Maxim., and 1 *T. rosthornii* Harms germplasm, which represented the main producing areas in China (Figure 1). The collected seed germplasm were subsequently used for research and preserved at the Institute of Bioresources and Applied Technology, Tongji University, China. The codes, locations, geographic coordinates (longitudes, latitudes), and species are detailed in Table 1.

### 2.2. Seed Morphological Characteristics Analysis

Seeds’ qualitative characteristics were observed and recorded, including their color, elliptical zones with uneven edges, the presence of linear ridges on the surface, and the flatness of the seed bottom [1] (Figure 2A). Quantitative descriptors included the seed length, width, thickness, and the weight of 100 seeds. For each germplasm, the length, width, and thickness of 10 randomly selected seeds were measured using a digimatic caliper (0 mm–150 mm, Shanghai Measuring & Cutting Tool Works Co., Ltd., Shanghai, China). The weight of 100 seeds was determined using a scale with 0.0001 g sensitivity.

### 2.3. Seed Microscopic Characteristics Analysis

The surface ornamentation of the seed coat was observed using scanning electron microscopy (SEM S-3400N Hitachi Science Systems Ltd., Hitachinaka, Japan). After sample preparation [16], SEM scanning was conducted at an accelerating voltage of 12.0 keV. The terminology of the seed ornamentation description was based on a previous study [17]. The width of the reticulum lines and the mesh area on the seed coat (Figure 2B) were quantitatively analyzed using ImageJ software (https://imagej.nih.gov/ij/, accessed on 10 April 2022).

### 2.4. DNA Extraction and PCR Amplification

The seeds were germinated in a light environment at room temperature until the seedlings reached a height of 3 cm–4 cm. Total genomic DNA was then extracted from the seedlings using the modified CTAB (cetyltrimethyl ammonium bromide) method [18]. The upstream primer (ITS-P17 5′-CTACCGATTGAATGGTCCGGTGAA-3′) and downstream primer (ITS-26S-82R 5′-TCCCGGTTCGCTCGCCGTTACTA-3′) [19] used for amplifying nrDNA-ITS sequences were synthesized by Shanghai Sangon Biological Engineering Technology and Service Co. (Shanghai, China). Amplifications were performed in a Mastercycler Gradient PCR Machine (Eppendorf, Germany) using the following program: initial denaturation at 95 °C for 5 min; followed by 34 cycles of 94 °C for 30 s, 56 °C for 45 s, and 72 °C for 45 s; and a final extension at 72 °C for 10 min. The sequencing of the nrDNA-ITS region was performed by GENEWIZ CO., Ltd. (Suzhou, China). 

### 2.5. Sequence Alignment and Phylogenetic Analysis

Bidirectional sequences of the nrDNA-ITS region were assembled using BioEdit 7.1.11 software. The final assembled sequences were submitted to GenBank for accession numbers (Table 1). Then, the nrDNA-ITS sequences of *T. kirilowii* Maxim. (GQ845146.1), *T. rosthornii* Harms (GU059525.1), and *T. laceribractea* Hayata (OP872715.1) from GenBank were combined to construct the phylogenetic trees. *Sicyos angulatus* (DQ005999.1), which has the most similar nrDNA-ITS sequence to genus *Trichosanthes*, was used as an outgroup. All sequences were aligned using the Clustal W in Molecular Evolutionary Genetics Analysis (MEGA) 7.0 software (https://www.megasoftware.net, accessed on 10 April 2022). Evolutionary trees were constructed based on Maximum parsimony (MP). For satisfactory reliability in phylogenetic construction, the bootstrap value was set at 1000 replicates. 

### 2.6. Preparation of Seed Oils

The seed oils were extracted using the Soxhlet extraction method [20]. The seed kernels were dried at 60 °C for 30 min and then crushed with a pulverizer. The ground seed kernel was wrapped in filter paper and placed in a Soxhlet extractor and then extracted with petroleum ether at 90 °C for 4 h. The solvent was evaporated using a rotary vacuum evaporator (Buchi R3, Flawil, Switzerland) at 80 °C. The mass of the obtained oil was recorded, and the yield of the seed oil was calculated. 

### 2.7. Fatty Acid Composition Analysis

The fatty acid composition of the extracted seed oil was analyzed using gas chromatograph (Agilent GC-6890, Los Angeles, CA, USA) equipped with a flame-ionization detector (FID) and a capillary column (DB-FFAP, 100 m × 0.25 mm, film thickness 0.20 µm) according to the National Standard of the People’s Republic of China GB5009.168-2016 [21]. The GC settings were as follows: nitrogen was used as a carrier gas with a flow rate of 1 mL/min and a split ratio of 1:100. The temperatures of the injection block and the detector were 270 °C and 280 °C, respectively. The column temperature was maintained at 100 °C for 13 min; then, it was programmed to increase to 180 °C at a rate of 10 °C/min, maintained at 180 °C for 6 min, raised to 200 °C at a rate of 1 °C/min, and maintained at 200 °C for 20 min; finally, it was raised to 230 °C at a rate of 4 °C/min and maintained at 230 °C for 10.5 min. GC chromatogram peaks were identified by comparing the retention time from the mixture of the standard FAME (fatty acid methyl esters, 37-component FAME Mix from Sigma-Aldrich, St. Louis, MO, USA). Each fatty acid peak was quantified using an internal standard concentration equivalent. Fatty acids were expressed as percentage of the total fatty acids content. Each sample was independently analyzed in three replicates.

### 2.8. C. elegans Assays

#### 2.8.1. Antioxidant Activity Assay

The antioxidant capabilities of the seed oil in *C. elegans* (N2, wild-type) were assessed according to the previously described method, with some slight modifications [22]. Briefly, the synchronized L4 worms were treated with 200 μg/mL of the extracted oil at 20 °C for 3 days. While untreated worms were kept as a blank control, equal DMSO and linseed oil were used as solvent control groups and positive control groups, respectively. After 3-day treatment, 20 worms from each treatment were exposed to 200 mM of paraquat on NGM (Nematode growth medium) plates containing 5-fluorodeoxyuridine (FUDR) to inhibit hatching. Mortality was recorded every 2 h over the duration of the experiment, and each experiment was conducted in triplicate.

#### 2.8.2. Lifespan Assay

To evaluate anti-aging activity, five germplasm were selected based on their antioxidant activities and fatty acid compositions, including ZJQT and SXHZ of *T. laceribractea* Hayata, SDJN and YNHH of *T. kirilowii* Maxim., and GXYL of *T. rosthornii* Harms. The lifespan assays were conducted with slight adjustments to the protocol [23], involving the treatment of synchronized L4 worms with 200 μg/mL of seed oil at 20 °C. Linseed oil served as the positive control, while untreated worms were used as the control group. Worm survival was monitored every 48 h, with a worm being declared dead if it did not respond to a gentle touch using a worm picker. The experiment was conducted with three independent replicates. Survival data were calculated using the Kaplan–Meier method, and differences in survival were analyzed using the log-rank test. The mean lifespan (MLS) was estimated using the following formula:$$MLS=\frac{1}{N}\sum_{j}\frac{x_{j}+x_{j+1}}{2}d_{j}$$
where dj denotes the number of worms that died in the age interval (x_j_ to x_j+1_) and *N* denotes the overall quantity of worms.

#### 2.8.3. Lipofuscin Assay

The intestinal autofluorescence of lipofuscin was assessed as previously described [24]. The synchronized L4 worms were transferred to treatment plates with 200 μg/mL of seed oil for 3 days and then washed and mounted on 2% agar pads for observation under a fluorescence microscope (Revolve FL, Echo Laboratories, San Diego, CA, USA). Images acquired with a 20× objective lens were processed using ImageJ software to quantify the average pixel intensity values.

#### 2.8.4. Healthspan Assay

The healthspan assay, which encompasses body bending, pharyngeal pumping, and fertility, was evaluated in accordance with the method described in reference [25], with minor modifications. The synchronized L4 worms were transferred to treatment plates with 200 μg/mL of seed oil. The number of instances of body bending in 30 s and pharyngeal pumping in 15 s were recorded on the 3rd and 7th days following treatment. In fertility assays, after the treatment with 200 μg/mL of seed oil from the L1 stage, individual L4 worms were isolated on separate NGM plates (without FUDR) for egg-laying, and then the worms were moved daily throughout their fertility period to prevent mixing with their offspring. Subsequently, the offspring from each worm were counted post-hatching. Linseed oil served as a positive control, while untreated worms were used as the control group. Each test was replicated three times, with a minimum of 30 worms per trail. 

### 2.9. Data Analysis

All values were presented as the mean ± standard deviation (SD) based on a minimum of three biological replicates. Graphs were generated using GraphPad Primer 8. Statistical analyses were conducted using SPSS 19.0 software, followed by one-way analysis of variance (ANOVA) for comparison between groups. SPSS 19.0 was also employed for correlation tests between antioxidant activity and fatty acid content. The Euclidean distances between different edible *Trichosanthes* germplasm were determined through the between-groups linkage method in SPSS 19.0, considering seed morphological and microscopic characteristics, fatty acid content, and antioxidant activities of the seed oils. Nei’s genetic distances based on nrDNA-ITS sequences were calculated using MEGA 7.0. The correlations between these distance matrices were investigated using the Mantel test in GenAlEx 6.5.

## 3. Results

### 3.1. Species Classification of Edible Trichosanthes Germplasm

The 18 edible *Trichosanthes* germplasm from *T. laceribractea* Hayata, *T. rosthornii* Harms, and *T. kirilowii* Maxim. revealed significant variances in their seed morphological and microscopic characteristics (Table 2 and Figure 3). All seeds exhibited varying shades of brown, with distinct morphological features, such as elliptical zones with irregular edges and pronounced linear ridges on the seed coat for all six *T. laceribractea* Hayata germplasm. In contrast, only three *T. kirilowii* Maxim. germplasm, specifically AHAQ, YNHH, and HNXX, displayed obvious linear ridges. The flat umbilicus-end of the seed was a key characteristic of *T. rosthornii* Harms germplasm GXYL. The quantitative characteristics of all seeds were not dependent on species. Among all germplasm, YNHH had the largest seed length (19.09 mm) and seed width (11.73 mm), whereas the smallest measurements were observed in HNLY (13.29 mm for length) and GZGY (7.27 mm for width), respectively. ZJQT displayed the greatest seed thickness at 5.57 mm, while the lowest thickness was noted in HNXX, measuring 3.58 mm. HNZZ had the heaviest 100-seed weight at 37.83 g, with the lowest recorded in HNXX at 18.83 g. For the microscopic characteristic, the seed coat surface decorations of all germplasm were of a reticulate type, exhibiting significant differences in the width of the reticulum lines and mesh area (Table 2 and Appendix A). 

The 18 edible *Trichosanthes* germplasm were categorized into three clusters according to their seed morphological and microscopic characteristics. However, these clusters did not align with the species classification (Figure 4A,B). Based on the combined seed morphological and microscopic characteristics, the germplasm were also grouped into three clusters, and this grouping was consistent with their species classification. Cluster I included all 11 *T. kirilowii* Maxim germplasm, Cluster II encompassed all 6 *T. laceribractea* Hayata germplasm, and the single *T. rosthornii* Harms germplasm GXYL was placed in Cluster III (Figure 4C). This finding revealed that the microscopic characteristics of the seed coat can serve as one of the indicators for species classification within *Trichosanthes* germplasm.

In the phylogenetic tree, the 18 edible *Trichosanthes* germplasm were divided into three clusters based on nrDNA-ITS sequences, with Sicyos angulatus (DQ005999.1) serving as the outgroup. Cluster I comprised all 11 *T. kirilowii* Maxim germplasm, Cluster II contained the *T. rosthornii* Harms germplasm GXYL, and Cluster III included all 6 *T. laceribractea* Hayata germplasm (Figure 4D). The Mantel test revealed a significant correlation (r = 0.577, *p* < 0.05) between Nei’s genetic distances based on nrDNA-ITS sequences and the combined morphology and microscopic characteristics of the 18 edible *Trichosanthes* germplasm. This suggests that the three *Trichosanthes* L. species can be identified and distinguished through either their seed morphological and microscopic characteristics or nrDNA-ITS sequences.

### 3.2. Yield and Fatty Acid Composition of Seed Oil

The seed oil yields and fatty acid compositions varied significantly both within and between species among the 18 edible *Trichosanthes* germplasm. The seed oil yields from *T. kirilowii* Maxim. (44.24–55.61%) and *T. laceribractea* Hayata (44.29–54.23%) were significantly higher than that of *T. rosthornii* Harms (42.28%) (Table 3). The primary five fatty acids were generally found in the following order of abundance: linoleic acid (LA, 29.82–40.40%), trichosanic acid (TA, 18.32–49.19%), oleic acid (OA, 6.22–42.58%), palmitic acid (PA, 3.20–8.20%), and stearic acid (SA, 2.60–4.71%) (Table 3). The total UFA content in *T. kirilowii* Maxim. (91.20–92.75%) and *T. rosthornii* Harms (91.50%) was significantly greater than that in *T. laceribractea* Hayata (87.38–89.65%). The fatty acid composition of seed oil allowed for the division of the 18 edible *Trichosanthes* germplasm into three clusters (Figure 4E). The pattern of this grouping demonstrates a significant positive correlation with the clusters based on combined seed morphological and microscopic characteristics (r = 0.242, *p* < 0.05), as well as with those based on nrDNA-ITS sequences (r = 0.337, *p* < 0.05). Cluster II included the *T. laceribractea* Hayata germplasm, which was primarily distinguished by low levels of UFAs, TA (25.54–29.99%), and high levels of PA (6.57–8.20%). In contrast, the germplasm of *T. kirilowii* Maxim. (Cluster I) and *T. rosthornii* Harms (Cluster III) show opposite characteristics (Table 3).

### 3.3. Antioxidant Activity of Seed Oil 

The seed oil from the 18 edible *Trichosanthes* germplasm significantly improved oxidative stress resistance in *C. elegans*. Following exposure to paraquat, the average lifespan of worms treated with seed oil varied from 18.09 h (FJNP) to 22.18 h (YNHH), showing significant enhancements compared to the blank controls (16.10 h) (Figure 5A and Appendix A). The percentage increase in mean lifespan compared to blank controls ranged from 12.36% (FJNP) to 37.76% (YNHH), whereas the positive control, linseed oil, resulted in a 29.69% increase (Appendix A). The antioxidant activity of seed oil from the 18 edible *Trichosanthes* germplasm exhibited a strong positive correlation with the total content of UFAs (r = 0.861, *p* < 0.01) and a strong negative correlation with the content of PA (r = −0.848, *p* < 0.01). There was no significant correlation (*p* > 0.05) with the content of SA (r= −0.352), OA (r = −0.242), LA (r = 0.166) or TA (r = 0.434). The Mantel test revealed a significant positive correlation (r = 0.337, *p* < 0.05) between the antioxidant activity of seed oil and Nei’s genetic distances based on the nrDNA-ITS sequences of the 18 edible *Trichosanthes* germplasm. This suggests that the antioxidant properties of seed oil in different edible *Trichosanthes* germplasm are species-specific. In general, the antioxidant activity of seed oil from *T. kirilowii* Maxim. and *T. rosthornii* Harms germplasm was stronger than that from the *T. laceribractea* Hayata germplasm. 

### 3.4. Anti-Aging Activity of Seed Oil

#### 3.4.1. Effect on Lifespan of *C. elegans*

The seed oil from the selected five germplasm, including ZJQT and SXHZ from *T. laceribractea* Hayata, SDJN and YNHH from *T. kirilowii* Maxim., and GXYL from *T. rosthornii* Harms, was able to significantly prolong the lifespan of *C. elegans*. The average lifespan of worms treated with these seed oils ranged from 17.59 d (ZJQT) to 22.17 d (SDJN), which was significantly longer than that of the blank controls (Figure 5B and Appendix A). The increase in mean lifespan due to seed oil treatment varied significantly, ranging from 15.40% (ZJQT) to 45.46% (SDJN) compared to the blank control. In contrast, the positive control, linseed oil, resulted in an 18.65% increase (Appendix A). Overall, the effect of seed oil from *T. kirilowii* Maxim. and *T. rosthornii* Harms on the lifespan of *C. elegans* was significantly more pronounced than that of *T. laceribractea* Hayata and the positive control, linseed oil. 

#### 3.4.2. Effect on Lipofuscin Accumulation of *C. elegans*

Lipofuscin is a complex mix comprising oxidized proteins and lipid degradation residues accompanied by smaller amounts of carbohydrates and metals. Intestinal autofluorescence in *C. elegans* indicates the accumulation of lipofuscin and served as a biomarker for cellular damage associated with aging [26]. Seed oil from the edible *Trichosanthes* germplasm, with the exception of SXHZ, was able to significantly diminish the accumulation of lipofuscin in worms (Figure 5C,G, Appendix A). This implies a potential delay in the aging process of *C. elegans*. 

#### 3.4.3. Effect on the Healthspan of *C. elegans*

The assessment of mobility and pharyngeal pumping rates, indicators of muscle health in *C. elegans*, revealed that the seed oil treatment significantly hindered the age-related deterioration in mobility. Worms treated with the seed oils from the edible *Trichosanthes* germplasm, with the exception of ZJQT, exhibited significantly higher body bending rates (Figure 5D) and pharyngeal pumping rates (Figure 5E) on both day 3 and day 7 after treatment (Appendix A). There was no significant difference in the progeny productions of worms between those treated with seed oil and the blank controls (Figure 5F). Therefore, the seed oil from edible *Trichosanthes* has been shown to significantly enhance muscle function and slow down the decline in physiological functions of worms, indicating its potential to improve the healthspan of *C. elegans*.

## 4. Discussion

*Trichosanthes* L. plants, known for their perennial herbaceous lianas with a dioecious system, exhibit considerable genetic diversity, attributed to their prevalent cross-pollination mechanism [4]. The genetic background of edible *Trichosanthes* germplasm is intricate, incorporating a range of species, cultivars, and landraces belonging to the *Trichosanthes* L. genus [27]. In this study, the 18 edible *Trichosanthes* germplasm from *T. laceribractea* Hayata, *T. rosthornii* Harms, and *T. kirilowii* Maxim. were differentiated at the species level based on seed morphological and microscopic characteristics, as well as nrDNA-ITS sequences. However, the quantitative characteristics of the seeds were not exclusively species-specific and could potentially be influenced by the growth conditions and environmental factors. The ornamentation of the seed coat is also an important feature used for microscopic identification and is rarely influenced by environmental factors [28]. Four types of seed coat ornamentation have been observed among 31 species and three variants of *Trichosanthes* L. using scanning electron microscopy in the reference [29]. In this study, the reticulate type of seed coat ornamentation was the only type observed across the 18 edible *Trichosanthes* germplasm belonging to three different species. The widths of the reticulum lines and the mesh area, as microscopic features of the seed coat, were used as morphology indicators to differentiate the three edible *Trichosanthes* species in conjunction with macroscopic traits of the seeds. Furthermore, our study indicates that nrDNA-ITS sequences varied significantly not only between *Trichosanthes* species, but also within individual species. The finding contradicts prior studies that found no variation in nrDNA-ITS sequences within species of *Trichosanthes* L. [27,30]. Consequently, it is essential to define a threshold for nrDNA-ITS sequence similarity to aid in the species identification of edible *Trichosanthes*.

The seed oil from edible *Trichosanthes* germplasm has the potential to boost the antioxidant capabilities of *C. elegans*. The antioxidant activities of the seed oils were specific to the species, with *T. kirilowii* Maxim. and *T. rosthornii* Harms exhibiting more potent activities compared to *T. laceribractea* Hayata. The 18 edible *Trichosanthes* germplasm were categorized into three consistent clusters based on the fatty acid composition of the seed oil and nrDNA-ITS sequences, respectively. This classification demonstrates a significant positive correlation (r = 0.337, *p* < 0.05). Plants with closely related genetic backgrounds tend to have similar chemical compositions [31], particularly regarding primary metabolites, like the fatty acid composition, which is primarily influenced by genetic factors [32]. The fatty acid composition of seed oil from *Trichosanthes* germplasm was also mainly determined by genetic factors, which in turn influence its antioxidant activity.

UFAs have been identified as the principal active components in the seed oil from *T. kirilowii* Maxim. [32]. In this study, the seed oils from *T. laceribractea* Hayata and *T. rosthornii* Harms exhibited a fatty acid composition similar to that of *T. kirilowii* Maxim., which was rich in UFAs, primarily including OA, LA, and TA. While there was no significant correlation between the antioxidant activity of seed oil and the content of any single unsaturated fatty acid, a significant correlation was observed between the total content of UFAs (r = 0.861, *p* < 0.01) and the ratio of trichosanic acid to oleic acid (TA/OA) (r = 0.475, *p* < 0.05) across the 18 edible *Trichosanthes* germplasm. In contrast, monounsaturated fatty acids, particularly OA, have been shown to significantly induce the production of ROS in mitochondria of cells [33]. Our previous study also revealed that the ratio of linoleic acid (LA) to oleic acid (OA) displayed a highly significant positive correlation with the inhibition rates in Job’s tears (*Coix lachryma-jobi* L.) seed oil for human bladder carcinoma T24 cells [34]. TA is an omega-5 isomer of conjugated α-linolenic acid (CLnA) known for its broad biological effects, including anti-inflammatory, anti-diabetic, anti-obesity, anti-proliferative, anti-carcinogenic, and antioxidant properties [35]. Seed oil exhibiting high antioxidant activity is characterized by a high content of total UFA content and a high ratio of TA/OA. Therefore, these factors can be considered important indicators for the quality control of seed oil from edible *Trichosanthes* germplasm.

Other seed oils rich in UFAs, such as linseed oil, have also been reported to exhibit antioxidant activity [36]. The selected five germplasm from three *Trichosanthes* species demonstrated that their seed oils could significantly extend the lifespan of *C. elegans*. However, in line with the antioxidant activity findings, the anti-aging efficacy of seed oil from *T. kirilowii* Maxim. and *T. rosthornii* Harms was comparable to that of linseed oil, but it was more potent than that from *T. laceribractea* Hayata. Notably, the seed oil from the SXHZ and ZJQT germplasm of *T. laceribractea* Hayata did not significantly impact the accumulation of lipofuscin, nor did it affect the mobility or pharyngeal pumping rates in *C. elegans*. Multiple studies have implied that interventions extending lifespan do not invariably promote healthspan [37,38]. The seed oil from *T. kirilowii* Maxim. and *T. rosthornii* Harms germplasm did more than just increase longevity; it also improved the healthspan of *C. elegans* by mitigating aging-related cellular damage and the decline of physiological functions. No significant difference was observed in reproduction between the experimental and control groups, indicating that the anti-aging effects of the seed oils might not be linked to the reproductive signaling pathway [39]. These findings suggest that the seed oil from edible *Trichosanthes* germplasm could be a potential natural antioxidant without associated toxicity.

The free radical theory of aging posits that the accumulation of ROS with age surpasses the organism’s ability to clear them, leading to cumulative oxidative stress that accelerates the aging process [40,41]. In line with this theory, it has been reported that the seed oil of *T. kirilowii* Maxim. exhibited in vitro antioxidant activity against the superoxide anion radical [13]. The anti-aging effect of seed oils from *Trichosanthes* germplasm is likely attributed to their antioxidant activity, which involves scavenging ROS. Nevertheless, the seed oil from the SDJN germplasm, despite its lower antioxidant activity, also exhibited a potent delaying effect on functional decline during aging. This suggests that the anti-aging effects of these seed oils are not solely mediated by their antioxidant activity. The mechanism behind the anti-aging effect of seed oil from edible *Trichosanthes* germplasm warrants further investigation. Our study highlighted how different edible *Trichosanthes* germplasm influence the antioxidant and anti-aging activity of their seed oil in *C. elegans*, indicating that edible *Trichosanthes* seed oil could be utilized as functional food to prolong lifespan and postpone aging. These insights set the stage for additional research into the various mechanisms at play and the wider application of these oils in the fields of health and nutrition.

## 5. Conclusions

The *Trichosanthes* germplasm from three species can be accurately identified and distinguished at the species level through their seed characteristics. Seed oils derived from various edible *Trichosanthes* germplasm significantly enhanced both antioxidant and anti-aging activities in *C. elegans*, with the seed oil from YNHH exhibiting the most pronounced effects. The observed biological activities of these seed oils are closely associated with their specific species and demonstrate a significant positive correlation with their total content of UFAs. Consequently, seeds from edible *Trichosanthes* germplasm emerge as promising functional foods, providing notable antioxidant and anti-aging benefits.

## Figures and Tables

**Figure 1 foods-13-00503-f001:**
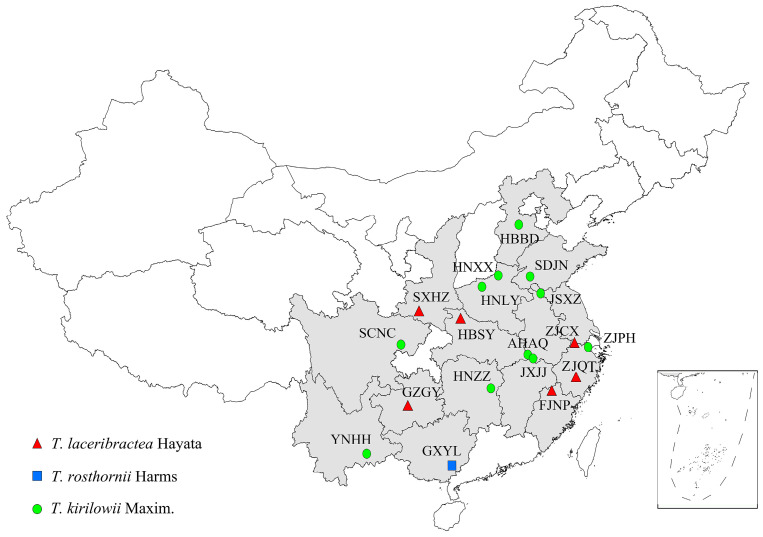
Geographic localities of 18 edible *Trichosanthes* germplasm used in this study. The germplasm codes are defined in Table 1.

**Figure 2 foods-13-00503-f002:**
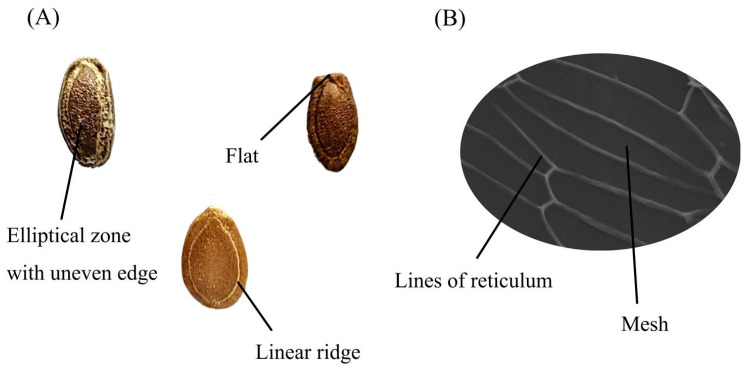
Seed qualitative morphological and microscopic characteristics of edible *Trichosanthes* germplasm. (**A**) Qualitative morphological characteristics (under naked-eye observation). (**B**) Surface decoration characteristics (under SEM, ×1000).

**Figure 3 foods-13-00503-f003:**
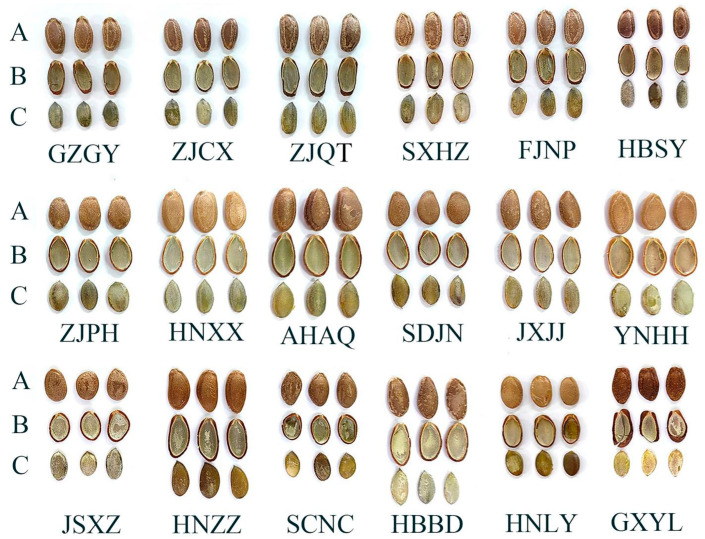
Seed morphological characteristics of 18 edible *Trichosanthes* germplasm. (**A**) Seed. (**B**) Seed hull. (**C**) Kernel. Seed color: chocolate brown (GZGY, ZJCX, ZJQT, SXHZ, FJNP, and HBSY); dark brown (AHAQ, HNZZ, and GXYL); brown (ZJPH, SDJN, JXJJ, JSXZ, SCNC, and HBDD); light brown (HNXX, YNHH, and HNLY). Seed coat surface with elliptical zone with uneven edge: GZGY, ZJCX, ZJQT, SXHZ, FJNP, and HBSY. Seed coat surface with obvious linear ridge: GZGY, ZJCX, ZJQT, SXHZ, FJNP, HBSY, HNXX, AHAQ, YNHH, and GXYL. The shape of the umbilicus-end of the seed was flat: GXYL.

**Figure 4 foods-13-00503-f004:**
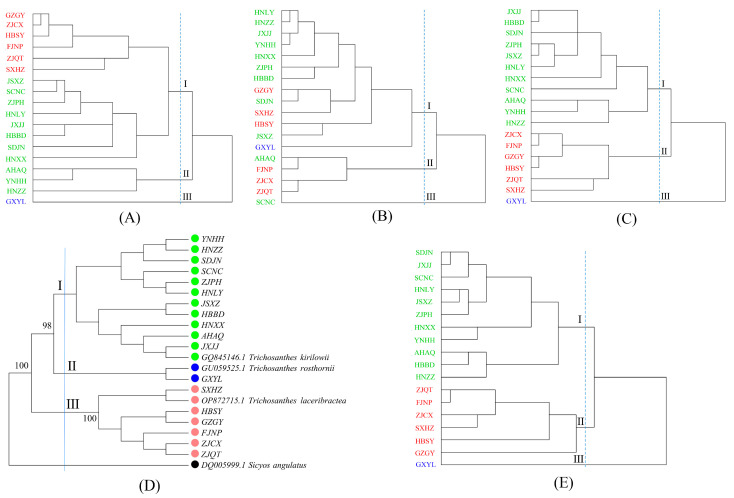
Cluster dendrogram of 18 edible *Trichosanthes* germplasm constructed using different indicators. (**A**) Seed morphological characteristics. (**B**) Seed microscopic characteristics. (**C**) Seed morphological and microscopic characteristics. (**D**) MP dendrogram based on nrDNA-ITS sequences of 18 germplasm and related species. The numbers between nodes indicate percentages of bootstrap support. (**E**) Fatty acid composition of seed oil. Red: *T. laceribractea* Hayata; Blue: *T. rosthornii* Harms; Green: *T. kirilowii* Maxim.

**Figure 5 foods-13-00503-f005:**
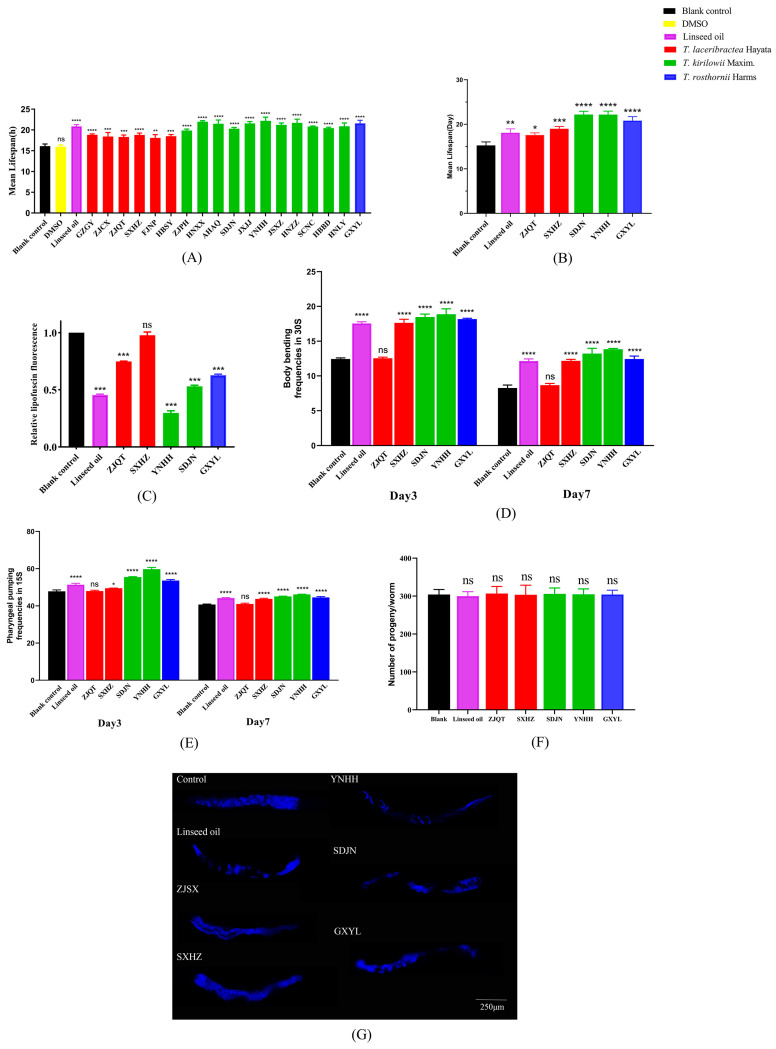
Effects of seed oil from different edible *Trichosanthes* germplasm on antioxidant and anti-aging activities in *C. elegans* N2 (wild-type). (**A**) Oxidative stress tolerance. (**B**) Longevity. (**C**) Relative lipofuscin fluorescence. (**D**) Body bending rate. (**E**) Pharyngeal pumping rate. (**F**) Number of offspring. (**G**) Lipofuscin accumulation (under FM, ×20). Data are presented as the mean ± SD of three independent experiments. * *p* < 0.05, ** *p* < 0.01, *** *p* < 0.001, **** *p* < 0.0001, ns: no significance vs. blank control.

**Table 1 foods-13-00503-t001:** Codes, geographic localities, and nrDNA-ITS sequence numbers of 18 edible *Trichosanthes* germplasm used in this study.

Code	Geographic Localities	Longitude (E) Latitude (N)	Species	GenBank Accession Number
GZGY	Guiyang, Guizhou Province	106°37′22.73″ (E) 26°40′43.14″ (N)	*T. laceribractea* Hayata	OR043672
ZJCX	Changxing, Zhejiang Province	119°54′36.40″ (E) 31°01′35.87″ (N)	*T. laceribractea* Hayata	OR043673
ZJQT	Qingtian, Zhejiang Province	120°17′22.38″(E) 28°08′23.53″(N)	*T. laceribractea* Hayata	OR043674
SXHZ	Hanzhong, Shanxi Province	107°01′54.98″ (E) 33°04′4.22″ (N)	*T. laceribractea* Hayata	OR043675
FJNP	Nanping, Fujian Province	118°07′13.54″ (E) 27°19′54.30″ (N)	*T. laceribractea* Hayata	OR043676
HBSY	Shiyan, Hubei Province	110°48′46.26″ (E) 32°35′30.30″ (N)	*T. laceribractea* Hayata	OR043677
GXYL	Yulin, Guangxi Province	110°03′4.50″ (E) 22°34′46.45″ (N)	*T. rosthornii* Harms	OR043666
ZJPH	Pinghu, Zhejiang Province	121°0′57.82″ (E) 30°40′33.06″ (N)	*T. kirilowii* Maxim.	OR043979
HNXX	Xinxiang, Henan Province	113°54′21.53″ (E) 35°22′18.48″ (N)	*T. kirilowii* Maxim.	OR043980
AHAQ	Anqing, Anhui Province	116°7′44.94″ (E) 30°9′12.96″ (N)	*T. kirilowii* Maxim.	OR043981
SDJN	Jining, Shandong Province	116°35′47.36″ (E) 35°24′29.52″ (N)	*T. kirilowii* Maxim.	OR043983
JXJJ	Jiujiang, Jiangxi Province	116°0′5.18″ (E) 29°42′19.80″ (N)	*T. kirilowii* Maxim.	OR043984
YNHH	Honghe, Yunnan Province	103°22′32.16″ (E) 23°21′51.19″ (N)	*T. kirilowii* Maxim.	OR043985
JSXZ	Xuzhou, Jiangsu Province	117°11′7.94″ (E) 34°17′17.63″ (N)	*T. kirilowii* Maxim.	OR043986
HNZZ	Zhuzhou, Hunan Province	113°10′27.19″ (E) 27°51′22.86″ (N)	*T. kirilowii* Maxim.	OR043987
SCNC	Nanchong, Sichuan Province	106°07′8.00″ (E) 30°46′53.50″ (N)	*T. kirilowii* Maxim.	OR043988
HBBD	Baoding, Hebei Province	115°27′31.50″ (E) 38°52′39.25″ (N)	*T. kirilowii* Maxim.	OR043989
HNLY	Luoyang, Henan Province	112°30′38.81″ (E) 34°42′15.52″ (N)	*T. kirilowii* Maxim.	OR043982

**Table 2 foods-13-00503-t002:** Seed morphological and microscopic characteristics of 18 edible *Trichosanthes* germplasm.

Code	Qualitative Characteristics				Quantitative Characteristics				Microscopic Characteristics	
	Color	Elliptical Zone with Uneven Edge	Obvious Linear Ridge	Umbilicus-End Is Flat	Length (mm)	Width (mm)	Thickness (mm)	100-Seed Weight (g)	Mesh Area (μm^2^)	Width of the Lines of the Reticulum (μm)
GZGY	Chocolate brown	Yes	Yes	No	14.11 ± 0.82 ^H–J^	7.27 ± 0.53 ^G^	4.68 ± 0.47 ^BC^	19.91 ± 0.42 ^KL^	1570.51 ± 39.24 ^E^	1.361 ± 0.054 ^H^
ZJCX	Chocolate brown	Yes	Yes	No	13.54 ± 0.97 ^IJ^	7.42 ± 0.82 ^G^	4.63 ± 0.44 ^C^	22.42 ± 0.33 ^HI^	2792.78 ± 51.86 ^B^	0.903 ± 0.007 ^I^
ZJQT	Chocolate brown	Yes	Yes	No	15.27 ± 1.85 ^FG^	8.45 ± 0.20 ^EF^	5.57 ± 0.48 ^A^	32.93 ± 0.39 ^B^	2950.39 ± 44.28 ^A^	0.803 ± 0.014 ^IJ^
SXHZ	Chocolate brown	Yes	Yes	No	16.00 ± 0.79 ^EF^	8.57 ± 0.51 ^EF^	5.40 ± 1.03 ^AB^	27.64 ± 0.51 ^C–E^	1601.48 ± 35.5 ^E^	0.841 ± 0.065 ^IJ^
FJNP	Chocolate brown	Yes	Yes	No	14.67 ± 0.68 ^G–I^	8.52 ± 0.66 ^EF^	4.65 ± 0.58 ^BC^	26.90 ± 0.43 ^DE^	2291.83 ± 118.30 ^C^	0.631 ± 0.048 ^K^
HBSY	Chocolate brown	Yes	Yes	No	14.26 ± 1.47 ^H-J^	8.20 ± 1.47 ^FG^	4.69 ± 1.01 ^BC^	20.03 ± 0.57 ^J-L^	1882.51 ± 85.11 ^D^	2.593 ± 0.098 ^C^
GXYL	Dark brown	No	Yes	Yes	14.72 ± 1.30 ^F–H^	9.90 ± 1.04 ^B–D^	3.82 ± 0.50 ^DE^	21.18 ± 0.46 ^I–K^	517.89 ± 22.04 ^I^	3.074 ± 0.037 ^B^
AHAQ	Dark brown	No	Yes	No	18.38 ± 1.55 ^AB^	10.64 ± 0.65 ^B^	4.72 ± 0.66 ^BC^	28.06 ± 0.76 ^CD^	2161.85 ± 113.66 ^C^	0.729 ± 0.053 ^JK^
HNZZ	Dark brown	No	No	No	18.21 ± 0.99 ^AB^	9.92 ± 0.66 ^B–D^	4.41 ± 0.23 ^CD^	37.83 ± 0.45 ^A^	1020.60 ± 43.68 ^FG^	1.642 ± 0.052 ^EF^
ZJPH	Brown	No	No	No	14.71 ± 0.85 ^G–I^	10.18 ± 1.00 ^BC^	4.47 ± 0.38 ^CD^	20.62 ± 0.33 ^I–L^	1140.17 ± 88.12 ^F^	2.035 ± 0.079 ^D^
SDJN	Brown	No	No	No	16.44 ± 0.76 ^D–F^	9.05 ± 0.57 ^D–F^	4.35 ± 0.74 ^CD^	21.38 ± 0.49 ^I–K^	1488.44 ± 14.78 ^E^	1.516 ± 0.041 ^FG^
JXJJ	Brown	No	No	No	17.67 ± 0.79 ^B–D^	9.26 ± 0.48 ^C–E^	4.17 ± 0.58 ^C–E^	23.79 ± 0.37 ^GH^	881.65 ± 24.51 ^G^	1.656 ± 0.033 ^E^
JSXZ	Brown	No	No	No	13.90 ± 0.93 ^IJ^	9.04 ± 0.70 ^D–F^	4.15 ± 0.67 ^C–E^	21.93 ± 1.79 ^H–J^	1875.18 ± 54.52 ^D^	2.024 ± 0.022 ^D^
SCNC	Brown	No	No	No	14.06 ± 1.15 ^H–J^	9.33 ± 0.97 ^C–E^	4.60 ± 0.45 ^C^	24.52 ± 1.54 ^FG^	1804.22 ± 14.90 ^D^	4.240 ± 0.086 ^A^
HBBD	Brown	No	No	No	17.83 ± 0.57 ^A–C^	9.67 ± 0.97 ^CD^	3.73 ± 0.74 ^DE^	19.14 ± 0.65 ^GH^	1458.33 ± 52.88 ^E^	2.152 ± 0.091 ^D^
YNHH	Light brown	No	Yes	No	19.09 ± 1.39 ^A^	11.73 ± 0.93 ^A^	4.37 ± 0.61 ^CD^	28.87 ± 0.43 ^C^	910.82 ± 9.61 ^G^	1.655 ± 0.025 ^EF^
HNLY	Light brown	No	No	No	13.29 ± 0.93 ^J^	8.40 ± 0.27 ^EF^	4.45 ± 0.43 ^CD^	26.04 ± 0.28 ^EF^	1014.93 ± 45.09 ^FG^	1.626 ± 0.048 ^E–G^
HNXX	Light brown	No	Yes	No	16.72 ± 1.50 ^C–E^	9.59 ± 0.68 ^CD^	3.58 ± 0.75 ^E^	18.83 ± 0.32 ^L^	691.58 ± 45.29 ^H^	1.494 ± 0.027 ^GH^

Different uppercases indicate significant difference at the level of 0.01 according to the Least-Significant Difference Test (LSD). Red: *T. laceribractea* Hayata; Blue: *T. rosthornii* Harms; Green: *T. kirilowii* Maxim.

**Table 3 foods-13-00503-t003:** Oil yield and main fatty acid composition of 18 edible *Trichosanthes* germplasm.

Code	Oil Yield (%)	UFA (%)	PA (%)	SA (%)	OA (%)	LA (%)	TA (%)
YNHH	48.88 ± 0.34 ^DE^	92.75 ± 0.16 ^A^	4.17 ± 0.09 ^DEF^	2.60 ± 0.06 ^I^	6.98 ± 0.19 ^K^	39.46 ± 0.23 ^AB^	45.53 ± 0.59 ^A^
JSXZ	54.82 ± 0.27 ^AB^	92.73 ± 0.29 ^A^	3.76 ± 0.15 ^FGH^	2.87 ± 0.11 ^GH^	16.48 ± 0.64 ^J^	37.94 ± 0.99 ^BCD^	37.68 ± 1.96 ^B^
JXJJ	47.53 ± 0.88 ^FG^	92.59 ± 0.47 ^A^	4.01 ± 0.48 ^EFG^	2.79 ± 0.03 ^HI^	22.39 ± 1.16 ^G^	34.67 ± 0.27 ^FGH^	34.79 ± 1.38 ^BCD^
HNLY	48.37 ± 0.07 ^D–F^	92.58 ± 0.05 ^A^	3.78 ± 0.04 ^FGH^	3.07 ± 0.01 ^FG^	24.70 ± 0.21 ^F^	34.87 ± 0.04 ^FG^	32.50 ± 0.21 ^DEF^
HNXX	45.12 ± 0.36 ^H^	92.55 ± 0.06 ^AB^	4.22 ± 0.03 ^DEF^	2.73 ± 0.02 ^HI^	6.22 ± 0.06 ^K^	35.88 ± 0.09 ^EF^	49.19 ± 0.28 ^A^
AHAQ	47.88 ± 0.68 ^E–G^	92.19 ± 0.05 ^ABC^	3.20 ± 0.04 ^H^	4.02 ± 0.04 ^BC^	32.43 ± 0.27 ^B^	33.30 ± 0.17 ^GHI^	25.88 ± 0.41 ^GH^
ZJPH	48.31 ± 0.65 ^D–G^	91.94 ± 0.05 ^A–D^	4.11 ± 0.03 ^D–G^	3.42 ± 0.02 ^DE^	20.42 ± 0.19 ^H^	37.04 ± 0.02 ^CDE^	33.63 ± 0.14 ^B–E^
SCNC	47.42 ± 0.38 ^FG^	91.67 ± 0.22 ^BCD^	4.48 ± 0.11 ^DE^	3.22 ± 0.11 ^DEF^	17.29 ± 0.56 ^IJ^	36.32 ± 0.22 ^DEF^	37.22 ± 0.44 ^BC^
HBBD	44.24 ± 0.50 ^H^	91.62 ± 0.04 ^CD^	3.84 ± 0.01 ^FG^	3.99 ± 0.03 ^BC^	28.70 ± 0.25 ^CD^	33.66 ± 0.07 ^GHI^	28.63 ± 0.37 ^FGH^
SDJN	55.61 ± 0.66 ^A^	91.60 ± 0.11 ^CD^	4.45 ± 0.06 ^DE^	3.45 ± 0.04 ^D^	19.07 ± 0.11 ^HI^	38.55 ± 0.12 ^BC^	33.31 ± 0.13 ^CDE^
GXYL	42.28 ± 0.48 ^I^	91.50 ± 0.05 ^CD^	4.70 ± 0.03 ^D^	3.17 ± 0.02 ^EF^	16.48 ± 0.08 ^J^	37.78 ± 0.18 ^BCD^	37.01 ± 0.32 ^BC^
HNZZ	49.33 ± 0.36 ^D^	91.20 ± 0.05 ^D^	3.52 ± 0.02 ^GH^	4.71 ± 0.03 ^A^	42.58 ± 0.31 ^A^	29.82 ± 0.17 ^J^	18.32 ± 0.21 ^I^
ZJCX	48.89 ± 0.54 ^DE^	89.65 ± 0.26 ^E^	6.90 ± 0.18 ^BC^	3.09 ± 0.06 ^FG^	29.13 ± 0.36 ^C^	32.92 ± 0.44 ^IH^	26.72 ± 1.16 ^GH^
ZJQT	44.92 ± 0.44 ^H^	89.65 ± 0.10 ^E^	6.57 ± 0.05 ^C^	3.44 ± 0.05 ^DE^	26.30 ± 0.09 ^EF^	33.08 ± 0.01 I^H^	29.58 ± 0.26 ^E–H^
GZGY	44.29 ± 0.49 ^H^	89.24 ± 0.08 ^E^	6.58 ± 0.05 ^C^	3.80 ± 0.03 ^C^	22.64 ± 0.17 ^G^	32.62 ± 0.24 ^IH^	33.66 ± 0.50 ^B–E^
SXHZ	52.39 ± 0.45 ^C^	89.16 ± 0.97 ^E^	7.25 ± 0.64 ^B^	3.24 ± 0.29 ^DEF^	26.25 ± 2.15 ^EF^	36.07 ± 2.21 ^EF^	25.54 ± 5.23 ^H^
FJNP	54.23 ± 0.36 ^B^	87.97 ± 0.76 ^F^	7.46 ± 0.49 ^B^	4.21 ± 0.24 ^B^	27.07 ± 1.55 ^DE^	33.13 ± 0.90 ^IH^	26.95 ± 3.22 ^GH^
HBSY	47.08 ± 0.41 ^G^	87.38 ± 0.43 ^F^	8.20 ± 0.25 ^A^	4.03 ± 0.16 ^BC^	15.81 ± 0.63 ^J^	40.40 ± 0.87 ^A^	29.99 ± 1.67 ^EFG^

Different uppercases indicate significant difference at the level of 0.01 according to the Least-Significant Difference Test (LSD). UFA: unsaturated fatty acid; PA: palmitic acid; SA: stearic acid; OA: oleic acid; LA: linoleic acid; TA: trichosanic acid. Red: *T. laceribractea* Hayata; Blue: *T. rosthornii* Harms; Green: *T. kirilowii* Maxim.

## Data Availability

All data have been made publicly available through the repository or in the Appendix A. The names of the repository and accession numbers can be found in the article.

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
