# Peer review of "Effect of Different Edible Trichosanthes Germplasm on Its Seed Oil to Enhance Antioxidant and Anti-Aging Activity in Caenorhabditis elegans"

_foods, 2024, doi:10.3390/foods13030503_

Round 1

Reviewer 1 Report

Comments and Suggestions for Authors

The reactive oxygen species produced during the metabolic process is one of the main causes of aging. There is a growing need to find compounds with antioxidant properties, low toxicity and minimal side effects to combat aging. Seed oil from edible Trichosanthes L. plants shows potential in this area. The study’s objectives were (i) to reveal the species classifications of edible Trichosanthes germplasm based on seed morphological and microscopic characteristics combined with nrDNA-ITS sequence; (ii) to analyze the yield and fatty acid composition of seed oil from the edible Trichosanthes germplasm; (iii) to evaluate the antioxidant and anti-aging properties of the seed oil using C. elegans

Table 2: Please correct the typing error… umbilicus-end is falt, check the typing errors in the whole manuscript

Table 3: please specify the acronyms UFA (%) PA (%) SA (%) OA (%) LA (%) TA (%) under the Table, for a better understanding.

Conclusion: please mention which of the 18  Trichosanthes germplasm was identified as superior in terms of antioxidant and antiaging activity in Caenorhabditis elegans.

Comments on the Quality of English Language

Minor editing of English language required and check the typing error in the whole manuscript

Reviewer 2 Report

Comments and Suggestions for Authors

In this study, the authors analysed seeds of three species of Trichosanthes germplasm focusing to the potential biological activities of their seed oil. Moreover, they were identified using a combination of seed morphological and microscopic characteristics and nrDNA-ITS sequences. Seed oils demonstrated to have antioxidant capacity related to its total unsaturated fatty acid content. In my opinion, the manuscript can be interesting for the field and the methodology applied was appropriate to reach the objectives proposed. However, I have some doubts and suggestions to authors. 

Figure 1 is mentioned in text before Table 1, so their position in the manuscript has to be interchanged.

Which does surface decoration characteristics mean?

Authors identified different shades of brown for the samples? How did they do it?

Why can be the morphological and microscopic characteristics not consistent with their species classification, however they are consistent based on the combined ones?

Is the value of r = 0.577 indicative of a good correlation between the Nei’s genetic distances or r = 0.337 or nrDNA-ITS sequences and r = 0.242 on the seed morphological combined microscopic characteristics? I have doubts with these correlations.

How do the authors think to study the mechanism of anti-aging effect of seed oil from edible Trichosanthes germplasm? 

In conclusion, could the authors propose which is the best Trichosanthes specie to follow the study?

Caption for Figure 5 includes results. It must be revised.

Authors must revise the captions of figures because of presence of white spaces between words. Also, they can check if all references (more than 5 years) are necessary.

Comments on the Quality of English Language

Correct text for spelling mistakes (senstivity, for example) 

Reviewer 3 Report

Comments and Suggestions for Authors

What do the PA (%), SA (%), OA (%), LA (%), TA (%) mean in the Table 3?

These abbreviations definitely need an explanation.

What does the TA/OA means (in 518 line)?

The abbreviation should be explain when appers in the text for a first time.

Reviewer 4 Report

Comments and Suggestions for Authors

The manuscript was to investigate the effect of edible Trichosanthes germplasm on its seed oil to enhance antioxidant and anti-aging activity. Some experiment design were conducted, but the authors should consider to revise the manuscript.

1. As far as we concerned, some enzyme inhibition studies focus on explicating the possible relationship between enzyme activity and various aging-related diseases. The authors should consider add necessary experiments to discuss the anti-aging activity of seed oil. For example, anti-tyrosinase, and anti-elastase activity are recommended to discuss.

2. The authors should check the English language throughout the manuscript since some parts were not easy to read.

3. In the introduction, the authors should update the relevant references to emphasize the importance of the study.

Comments on the Quality of English Language

The authors should check the English language throughout the manuscript since some parts were not easy to read.

Round 2

Reviewer 2 Report

Comments and Suggestions for Authors

Dear, authors:

The manuscript have been improved, so, my decision is that it can be published.

Best regards